The value of nanocarbon contrast methylene blue based on dye-based tracer technology in sentinel lymph node biopsy for breast cancer: a systematic review and meta-analysis

Xia Luhua 1
Zou Zuowei 2
Chong Le 381594706@qq.com 3
Wang Xinhua 1
Dong Zhanfei 1
Zhao Yanping 1
1 Tumor Hospital Affiliated to Xinjiang Medical University, Key Laboratory of Oncology of Xinjiang Uyghur Autonomous Region Urumqi, Department of Nuclear Medicine , Urumqi , Xinjiang , China
2 National Cancer Center/National Clinical Research Center for Cancer/Cancer Hospital, Chinese Academy of Medical Sciences and Peking Union Medical College, Department of Nuclear Medicine , Beijing , China
3 The Affiliated Cancer Hospital of Xinjiang Medical University, Department of Ultrasonography , Urumqi , Xinjiang , China
Upadhyay Rohit
Electronic publication date: 2025 Jun 11
Publication date: 2025
Volume: 13
Electronic Location ID: e19546
Received 2025 Jan 13; Accepted 2025 May 12
Copyright: ©2025 Xia et al.
Copyright year: 2025
Copyright holder: Xia et al.
License: This is an open access article distributed under the terms of the Creative Commons Attribution License, which permits unrestricted use, distribution, reproduction and adaptation in any medium and for any purpose provided that it is properly attributed. For attribution, the original author(s), title, publication source (PeerJ) and either DOI or URL of the article must be cited.
License URL: https://creativecommons.org/licenses/by/4.0/

Keywords: Breast cancer, Nanocarbon suspension, Methylene blue, Diagnostic value

Funding: The Xinjiang Uygur Autonomous Region Natural Science Foundation No. 2023D01C134 Key Laboratory of Oncology of Xinjiang Uyghur Autonomous Region No. XJKLO 2023U005 This work was supported by the Xinjiang Uygur Autonomous Region Natural Science Foundation (No. 2023D01C134) and Key Laboratory of Oncology of Xinjiang Uyghur Autonomous Region (No. XJKLO 2023U005). There was no additional external funding received for this study. The funders had no role in study design, data collection and analysis, decision to publish, or preparation of the manuscript.

==============================
Objective

The purpose of this study was to compare the diagnostic value of nanocarbon suspensions and methylene blue injections in sentinel lymph node biopsies of patients with breast cancer based on the dye method.

Methods

A systematic search of the PubMed, Embase, Cochrane Library (Central) and Web of Science (SCI Expanded) databases was performed to determine the diagnostic value of carbon nanoparticles in Chinese databases (China National Knowledge Infrastructure (CNKI), VIP, Wanfang, Chinese Biomedical Literature Database (CBM)) for identifying methylene blue in sentinel lymph node biopsies of patients with breast cancer. The QUADAS2 quality evaluation tool of Review Manager 5.4 was used to evaluate the quality of the included studies. Meta-Disc 1.4 software was used to calculate the extracted valid data and perform a heterogeneity test. STATA 14.0 software was selected to conduct a sensitivity analysis, and Deek’s publication bias test was used for the included studies.

Results

In 17 articles, the binding sensitivity of nanocarbons was 0.93 (0.90–0.95), the binding specificity was 0.98 (0.97–0.99), the binding sensitivity of methylene blue was 0.89 (0.85–0.92), and the binding specificity was 0.94 (0.92–0.95). The AUC value of the nanocarbon SROC was 0.9827 (SE = 0.0062), and the AUC value of methylene blue was 0.9495 (SE = 0.0139).

Conclusion

Nanocarbon versus methylene blue is a more satisfactory dye tracer for sentinel lymph node biopsy in breast cancer patients and should be considered a first-line diagnostic agent.

Introduction

Breast cancer is the leading cause of cancer death in women worldwide and the second most common cause of cancer death in women in the United States (Swaminathan, Saravanamurali & Yadav, 2023; Loibl et al., 2021; Rabinski & Brawley, 2022; Katsura et al., 2022). Lymph node metastasis is the most common metastatic mode of breast cancer, and the sentinel lymph nodes (sentinel lymph nodes, SLNs) are the first station lymph nodes that must be passed by lymph node metastasis from the primary tumour. Systemic metastasis is often judged clinically based on their pathological examination results. Theoretically, if the sentinel lymph nodes of breast cancer patients have not undergone metastasis, other lymph nodes in the axillary lymphatic drainage area will not metastasize (Saidha, Aggarwal & Sen, 2018). Sentinel lymph node biopsy of breast cancer can be used to determine whether axillary lymph nodes have metastasized, mainly through a less invasive sentinel lymph node biopsy (SLNB), to determine the tumour stage, estimate the prognosis, and formulate a comprehensive treatment plan. Patients with negative sentinel lymph nodes can be exempt from axillary lymph node dissection (ALND), which reduces the possibility of upper limb lymphedema due to surgical damage to the axilla and disruption of the anatomical structure, narrows the scope of the surgery, reduces surgical trauma, and improves patients’ quality of life (Ozkurt et al., 2019; Okur et al., 2020; Mathelin & Lodi, 2021).

The selection of an appropriate tracer is the primary condition for sentinel lymph node biopsy (SLNB). The dye method is the most commonly used tracing approach, with tracers primarily including blue dye and nanocarbon. Methylene blue (MB) is the most commonly used material for the blue dye method (Aziz et al., 2023). Nanocarbon suspension (NCS) is the first new third-generation specific lymph node tracer approved by the State Food and Drug Administration (SFDA). It is generated using a patented preparation technology to grind activated carbon particles into small particles, with saline and polyvinylpyrrolidone (PVP) added to prepare a suspension (Fiorito et al., 2006), and it has been used as a lymph node tracer in surgery for thyroid, breast, gastric, cervical, and colorectal cancers. This lymph node tracer (Hao et al., 2012; Cai et al., 2012; Liu et al., 2013; Zhang et al., 2019; Yan et al., 2016; Lu et al., 2017; Wang et al., 2011; Gao et al., 2021; Lin et al., 2017) has been widely used in rectal cancer surgery and has achieved good tracer results.

We included a comparative study on the diagnostic value of NCS versus MBI in SLNB of breast cancer, a meta-analysis of the value application of these two dye-based methods in SLNB of breast cancer, and a systematic comparison of the accuracy and diagnostic value of NCS injection versus MBI in SLNB.

Materials and Methods

Study search strategy

We conducted a systematic literature search across PubMed, EMBASE, Web of Science, Cochrane, and Chinese databases (China National Knowledge Infrastructure (CNKI), VIP, Wanfang, Chinese Biomedical Literature Database (CBM)) from January 2010 to January 2024 to identify relevant articles. The literature searchers are Luhua Xia, Zuowei Zou, the arbiter is Le Chong. The method used is to combine the subject word with a free word. The following terms, without language restriction, were used: (‘carbon nanoparticle suspension’ OR ‘nanocarbon’ OR ‘nano carbon’ OR ‘Nanogate Carbon’ OR ‘Carbon Nanoparticles’ OR ‘Nano carbon suspension’) AND (‘Lymph Nodes, Sentinel’ OR ‘Lymph Node, Sentinel’ OR ‘Sentinel Lymph Node’ OR ‘Sentinel’ OR ‘Node, Sentinal’ OR ‘Node, Sentinal’ OR ‘Sentinel Gland’) AND (‘Breast Cancer’ OR ‘breast carcinoma’ OR ‘breast cancer’ OR ‘breast tumor’ OR ‘Breast Carcinoma’ OR ‘Breast Neoplasm’). In addition to database searches, reference lists of eligible articles were also screened to obtain additional relevant articles. The search strategy for Chinese journals was similar to that used for English journals. A PROSPERO registration number was obtained for this study: CRD42024527545.

Study selection

Inclusion criteria were as follows: (1) diagnostic study of the use of two tracers, NCS and MBI, in sentinel lymph node biopsy for breast cancer. (2) All case studies in the included literature are female. (3) Patients with pathological biopsy confirmation of breast cancer. (4) Examination of the axillary lymph nodes before surgery was negative. (5) No history of previous axillary surgery or breast radiotherapy. (6) Breast cancer patients who are not pregnant and breastfeeding. (7) Studies for which complete data can be obtained.

Exclusion criteria were as follows: (1) Duplicate studies. (2) Types of literature: case reports, reviews, letters, comments, and meta-analysis studies. (3) Experiments on animals. (4) Research with complete data in the 2 × 2 table cannot be obtained.

Data extraction

Two authors (Luhua Xia, Zuowei Zou) independently assessed each article and then downloaded and extracted all data via standard data abstraction tables. Extracted data included publication year, number of true-positives, number of false-positives, number of false-negatives, number of true-negatives, sensitivity, and specificity. A 2 × 2 contingency table was created for each study. We calculated sensitivity, specificity and likelihood ratio (LR).

Quality assessment

All included studies were independently assessed for methodological quality by two authors via the Quality Assessment of Diagnostic Accuracy Studies (QUADAS2) tool (Whiting et al., 2011). If both readers disagree, a third reader will judge. The literature seeker is Luhua Xia Zuowei Zou, the referee is Le Chong. None of the readers were involved in the included studies. The QUADAS-2 checklist consists of four domains: patient selection, index test, reference standard and flow/timing. Each of the four domains was assessed for risk of bias based on several questions.

Statistical analyses

Sensitivity, specificity, positive likelihood ratio, and negative likelihood ratio are extracted or calculated from studies and then pooled to assess diagnostic accuracy. Subject summary occupational characteristics (SROC) curves were constructed to assess diagnostic accuracy. Heterogeneity between studies was examined through the inconsistency index (I2) statistic, with I2 values greater than 50% considered significant (Higgins et al., 2003), and heterogeneous data were analyzed through random effects models and fixed effects models (Dersimonian & Laird, 1986). The ability of published methods to analyze the performance of nanocarbon versus methylene blue for diagnosing anterior lymph nodes was explored through sensitivity analysis. Deek’s bias test was used to assess research publication bias.

Results

Search results

The flowchart of the literature retrieval process is presented below. According to the search strategy, an initial total of 402 articles were identified. After excluding 215 duplicate articles, 23 review articles, case reports, systematic reviews, and meta-analyses, seven animal experiments, and 155 articles whose research content did not align with the study objectives after abstract screening, a further 10 articles were excluded after full-text reading due to non-relevant research content. Additionally, three articles were excluded due to poor experimental design rigor. Ultimately, 17 articles (Jiang, Cao & Zhou, 2021; Wu & Chen, 2019; Liu, Bao & Qiu, 2019; Xia, Cao & Yang, 2019; Li & Kong, 2018; Qi, 2018; Wang, Chen & Zhao, 2017; Zhu, Li & Zhang, 2017; Zhang, 2017; Chen, 2015; Fei, 2015; Wu et al., 2015; Sun, Ge & Cao, 2013; Zhou, Du & Chen, 2012; Lin, Zhou & Li, 2012; Jie, Yan & Cao, 2011; Yang & Li, 2011) were included in this study, with no additional eligible articles retrieved (Fig. 1).

Figure 1 The study selection process.

Study characteristics

Table 1 lists the detailed characteristics of these studies. The studies (1,171 patients in the nanocarbon group and 1,076 patients in the sub-A group) were conducted in China from 2011–2024, with 22–88 years. Sixteen publications were in Chinese, and one publication was in English. As NCS is a third-generation specific tracer approved by the Chinese Food and Drug Administration, it is currently marketed only within China and has not entered the international market; therefore, no relevant studies from other countries were retrieved.

Table 1 Baseline characteristics of the included studies.

First author	Published year	Country	Age	Nano carbon	Methylene blue	
				TP	FP	FN	TN	Total	TP	FP	FN	TN	Total	
Jie GE	2011	China	28–73	21	0	2	41	64	13	6	2	20	51	
Lijie YANG	2011	China	Unclear	11	2	1	28	42	12	3	1	28	44	
Yi ZHOU	2012	China	28–76	29	4	2	43	78	23	11	1	30	65	
Qimou LIN	2012	China	26–72	25	2	1	36	64	26	8	1	22	57	
Xuan SUN	2013	China	28–68	37	0	2	49	88	30	0	4	56	90	
Xiufeng WU	2015	China	24–72	24	0	3	56	83	16	0	3	54	73	
Pangzhou CHEN	2015	China	33–48	9	2	0	39	50	6	0	3	27	36	
Fei MAI	2015	China	27–69	19	0	2	22	43	14	0	1	16	31	
Lei WANG	2017	China	22–68	12	1	1	39	53	15	6	1	25	47	
Qin ZHU	2017	China	30–65	50	3	3	90	146	51	7	5	85	148	
Jingwen ZHANG	2017	China	Average 50.5	20	0	2	118	140	31	0	6	108	145	
Dihang LI	2018	China	24–71	15	0	1	31	47	14	0	1	28	43	
Xiao QI	2018	China	30–84	16	0	1	35	52	17	0	2	31	50	
Peiyang WU	2019	China	29–72	15	0	1	30	46	11	0	2	17	30	
Xiaomin LIU	2019	China	32–71	15	1	2	41	59	14	1	3	41	59	
Guangfa XIA	2019	China	25–82	25	1	1	59	86	11	1	1	64	77	
Ningxiang JIANG	2021	China	36–88	8	1	1	20	30	6	3	2	19	30	
Notes.

Jie, Yan & Cao (2011), Yang & Li (2011), Zhou, Du & Chen (2012), Lin, Zhou & Li (2012), Sun, Ge & Cao (2013), Wu et al. (2015), Chen (2015), Wang, Chen & Zhao (2017), Zhu, Li & Zhang (2017), Zhang (2017), Li et al. (2018), Qi (2018), Wu & Chen (2019), Liu, Bao & Qiu (2019), Xia, Cao & Yang (2019), Jiang, Cao & Zhou (2021).

Quality assessment

Assessment of the quality of the literature via the QUADAS2 quality assessment tool in the ReviewMan5.4 software revealed that all included studies were of relatively high quality. Regarding reference standards, none of the 17 studies mentioned whether blinding was used in the interpretation of gold standard results, so there are concerns regarding reference standards. However, there were no concerns regarding patient selection, index tests, or flow and timing. None of these studies was deemed necessary to be excluded from the meta-analysis based on QUADAS-2 scoring (Figs. 2 and 3).

Figure 2 Methodological quality graph.

Jie, Yan & Cao (2011), Yang & Li (2011), Zhou, Du & Chen (2012), Lin, Zhou & Li (2012), Sun, Ge & Cao (2013), Wu et al. (2015), Chen (2015), Wang, Chen & Zhao (2017), Zhu, Li & Zhang (2017), Zhang (2017), Li et al. (2018), Qi (2018), Wu & Chen (2019), Liu, Bao & Qiu (2019), Xia, Cao & Yang (2019), Jiang, Cao & Zhou (2021).

Figure 3 Methodological quality summary.

Meta-analysis results

NCS group

The graphs (Figs. 4–5) clearly show that the sensitivity I2 = 0 and specificity I2 = 40.9%, the combined sensitivity of the selected fixed effects model was 0.93 (0.90–0.95), the combined specificity was 0.98 (0.97–0.99), the combined positive likelihood ratio was 24.72 (16.33–38.41), the combined negative likelihood ratio was 0.09 (0.06–0.12), and the combined diagnostic advantage ratio DOR was 439.10 (232.50–829.29).

Figure 4 NCS pooled sensitivity.

Jie, Yan & Cao (2011), Yang & Li (2011), Zhou, Du & Chen (2012), Lin, Zhou & Li (2012), Sun, Ge & Cao (2013), Wu et al. (2015), Chen (2015), Wang, Chen & Zhao (2017), Zhu, Li & Zhang (2017), Zhang (2017), Li et al. (2018), Qi (2018), Wu & Chen (2019), Liu, Bao & Qiu (2019), Xia, Cao & Yang (2019), Jiang, Cao & Zhou (2021).

Figure 5 NCS pooled specificity.

Jie, Yan & Cao (2011), Yang & Li (2011), Zhou, Du & Chen (2012), Lin, Zhou & Li (2012), Sun, Ge & Cao (2013), Wu et al. (2015), Chen (2015), Wang, Chen & Zhao (2017), Zhu, Li & Zhang (2017), Zhang (2017), Li et al. (2018), Qi (2018), Wu & Chen (2019), Liu, Bao & Qiu (2019), Xia, Cao & Yang (2019), Jiang, Cao & Zhou (2021).

Figure 6 MBI pooled sensitivity.

Jie, Yan & Cao (2011), Yang & Li (2011), Zhou, Du & Chen (2012), Lin, Zhou & Li (2012), Sun, Ge & Cao (2013), Wu et al. (2015), Chen (2015), Wang, Chen & Zhao (2017), Zhu, Li & Zhang (2017), Zhang (2017), Li et al. (2018), Qi (2018), Wu & Chen (2019), Liu, Bao & Qiu (2019), Xia, Cao & Yang (2019), Jiang, Cao & Zhou (2021).

Figure 7 MBI pooled specificity.

Jie, Yan & Cao (2011), Yang & Li (2011), Zhou, Du & Chen (2012), Lin, Zhou & Li (2012), Sun, Ge & Cao (2013), Wu et al. (2015), Chen (2015), Wang, Chen & Zhao (2017), Zhu, Li & Zhang (2017), Zhang (2017), Li et al. (2018), Qi (2018), Wu & Chen (2019), Liu, Bao & Qiu (2019), Xia, Cao & Yang (2019), Jiang, Cao & Zhou (2021).

MBI group

The graphs (Figs. 6–7) clearly show that the sensitivity I2 = 0 and specificity I2 = 83%. The random effects model was chosen to have a pooled sensitivity of 0.89 (0.85–0.92), pooled specificity of 0.94 (0.92–0.95), pooled positive likelihood ratio of 12.87 (6.93–23.90), pooled negative likelihood ratio of 0.15 (0.12–0.20), and pooled diagnostic advantage ratio DOR of 117.07 (67.17–204.03).

The summary data table of the meta-analysis results can be found in Table 2.

SROC curve

Based on the SROC curves, AUC = 0.9827 in the NCS group and AUC = 0.9495 in the MBI group, the diagnostic accuracy of breast cancer sentinel lymph node biopsy was as high as 98.27% and 94.95%, respectively, when the NCS and MBI were used as tracers (Figs. 8–9).

Figure 8 SROC curve of the NCS.

Figure 9 SROC curve of the BMI.

Exploration of heterogeneity

Threshold effect heterogeneity (NCS group)

The current data were imported into MetaDiSc14.0 software for analysis, resulting in a nonsignificant Spearman correlation coefficient of 0.417 (P = 0.095 > 0.05) between the logarithm of sensitivity and the logarithm of (1-specificity). Since the p value of b was > 0.05, by plotting the symmetric SROC curve (Fig. 8), there was no “shoulder-arm shape”, suggesting that there was no heterogeneity caused by the threshold effect in this study.

Non-threshold effect heterogeneity (NCS group)

The Cochran-Q test for the diagnostic advantage ratio (DOR) yielded a Cochran-Q = 5.35, P = 0.9937 > 0.001, implying that there was no heterogeneity due to nonthreshold effects in this study.

Table 2 Summary table of meta-analysis results.

Group	Pooled	Combined effect (95% CI)	Chi 2 /Cochran-Q	HeterogeneityI 2 (%)	Effect of model	
Nano carbon	Sensitivity	0.93 (0.90–0.95)	4.46	0	Fixed	
	Specificity	0.98 (0.97–0.99)	27.09	40.9	Fixed	
	Positive LR	32.68 (21.40–49.89)	13.39	0	Fixed	
	Negative LR	0.08 (0.06–0.12)	3.77	0	Fixed	
	Diagnostic Odds Ratio	450.52 (240.70–843.25)	5.35	0	Fixed	
Methylene blue	Sensitivity	0.89 (0.85–0.92)	11.7	0	Fixed	
	Specificity	0.94 (0.92–0.95)	94.08	83	Random	
	Positive LR	12.87 (6.93–23.90)	64.51	75.2	Random	
	Negative LR	0.15 (0.12–0.20)	11.67	0	Fixed	
	Diagnostic Odds Ratio	117.07 (67.17–204.03)	15.96	0	Fixed	

Threshold effect heterogeneity (MBI group)

The current data were imported into MetaDiSc14 software for analysis, yielding a Spearman correlation coefficient of 0.432 (p = 0.084 > 0.05) between the logarithm of sensitivity and the logarithm of (1-specificity), which was not significant. Furthermore, because the p value for b was < 0.05, the choice was made to plot asymmetric SROC curves (Fig. 9), which did not appear to be “shoulder-arm shape”, suggesting that there was no heterogeneity due to threshold effects in this study.

Non-threshold effect heterogeneity (MBI group)

The Cochran-Q test for the diagnostic superiority ratio (DOR) yielded a Cochran-Q = 15.96, P = 0.4557 > 0.001, implying that there was no heterogeneity due to nonthreshold effects in this study.

Sensitivity analysis

STATA14.0 software was chosen to perform a sensitivity analysis of the data in this study, and it can be clearly seen that the 17 included studies in the NCS group (Fig. 10) and MBI group (Fig. 11) do not have strong sensitivities, corresponding to a lack of sensitivity of the arithmetic results and suggesting that the results of this study are relatively stable.

Figure 10 NCS sensitivity analysis.

Figure 11 MBI sensitivity analysis.

Publication bias

The Deeks funnel plot asymmetry test revealed that there was no publication bias between studies in the NCS group (P = 0.10 > 0.05) (Fig. 12) or between studies in the MBI group (P = 0.14 > 0.05) (Fig. 13).

Figure 12 Publication bias test of the NCS.

Figure 13 Publication bias test of the MBI.

Clinical utility

Figure 14 shows the likelihood nomograph used by Fagan NCS in SLNB. The prior probability is set to 20%. NCS is used as a tracer for SLNB in breast cancer. If the result is positive, the probability of metastasis to pre-SLNs is 93%. If the result is negative, the probability of SLN metastasis is 2%.

Figure 14 Probability of the Fagan likelihood nomograph and NCS in detecting lymph node metastasis by SLNB.

Figure 15 shows the likelihood nomograph used by the Fagan MBI in SLNB. The prior probability is set to 20%. MBI is used as a tracer for SLNB of breast cancer. If the result is positive, the probability of metastasis pre-MBI is 90%. If the result is negative, the probability of SLN metastasis is 3%.

Figure 15 Probability of the Fagan likelihood nomograph and MBI in detecting lymph node metastasis by SLNB.

Discussion

SLNB was performed by Giuliano et al. (1994); the use of SLNB dye as a tracer for breast cancer was first reported by Krag et al. (1998), and the radionuclide tracer method was used. At present, there are three main methods of the SLNB technique: the dye method, the radionuclide tracer (nuclide method) and the dye method combined with the nuclide method for detection. Although the dye method is simple to operate, intuitively practical, and cost-effective without the need for special equipment, it has certain blindness, is time-consuming, and causes relatively large tissue damage. The dye method, relatively simple in operation, has been widely applied, primarily using tracers such as MBI and NCS. Regarding blue dye tracers, China predominantly utilizes methylene blue, whereas other countries more frequently employ patent blue and isosulfan blue. The success rates and false-negative rates of different blue dyes used as tracers for sentinel lymph node biopsy (SLNB) are similar (Simmons, Smith & Osborne, 2001). The radionuclide method primarily uses Tc-labeled sulfur colloid, antimony colloid, and other tracers. The advantage of the radionuclide method lies in its ability to clearly define the location preoperatively, providing strong surgical targeting with minimal damage; however, it involves radioactive contamination, expensive equipment, and complex operation, which limits its clinical application and promotion. The combined method involves the joint use of dyes and radionuclides as tracers, featuring visual identification and precise radionuclide localization, achieving high accuracy and a low false-negative rate, thus gaining favor among many surgeons.

The selection of an appropriate tracer is the primary condition for the success of SLNB and should possess the following characteristics: (1) fast absorption by lymphatic tissue and ability to clearly show lymphatic vessels and lymph nodes; (2) ability to accumulate and stay in the SLN for a long time; and (3) low cost and nontoxicity. To compensate for the shortcomings of traditional tracers, several new tracers have been produced, such as NCS, indocyanine green fluorescence, microbubble contrast-enhanced ultrasound, and superparamagnetic iron oxide nanoparticles (Cui et al., 2013; Zhou et al., 2016; Nielsen et al., 2017; Man et al., 2019; Bove et al., 2021), which have achieved better tracer effects. Indocyanine green (ICG) as a tracer offers several advantages, including ease of use, real-time intraoperative navigation, and high success rates. However, the fluorescence emitted by the tracer has low penetration depth, making it difficult to detect sentinel lymph nodes that are located deeper. Additionally, during anatomical dissection, lymphatic vessels can be easily severed, leading to leakage of the fluorescent tracer and subsequent fluorescence contamination of surrounding tissues, which complicates the identification and localization of sentinel lymph nodes. Microbubble contrast-enhanced ultrasound provides the advantage of enhancing contrast between surrounding tissues, but its disadvantages include relatively lower success rates and higher false-negative rates, necessitating further support from evidence-based medicine for its clinical application. Superparamagnetic iron oxide (SPIO) also shows potential; however, its main drawbacks include the need to remove metallic retractors from the incision site when reading magnetic signals, substantial medical costs, and the risk of skin pigmentation in patients.

In a combined analysis of 17 studies (1,171 patients in the NCS group and 1,076 patients in the MBI group), the sensitivity and specificity of the NCS for diagnosing sentinel lymph nodes was 0.93 (0.90–0.95) and 0.98 (0.97–0.99), respectively, and the sensitivity and specificity of the MBI for diagnosing breast cancer was 0.89 (0.85–0.92) and 0.94 (0.92–0.95), respectively. The AUC value of the SROC in the NCS group (AUC = 0.9827, SE = 0.0062) was significantly greater than that in the MBI group (AUC = 0.9495, SE = 0.0139).

Diagnostic tests often have difficulty conforming to the principles of randomized controlled trials. In practice, it is almost impossible to randomize the “test” and “control” groups, and most diagnostic results are heterogeneous because different researchers often use different thresholds for determining test results. Among the 17 included studies, the NCS group did not show heterogeneity, and a fixed-effects model was chosen, whereas the MBI group showed heterogeneity; therefore, a random-effects model was used (Jackson, White & Thompson, 2010), which is an accepted practice to appropriately increase the weight of the small-sample data and decrease the weight of the large-sample data to cope with the heterogeneity of the data. However, this may introduce certain risks because the quality of the small-sample data is usually poorer and more biased, which may lead to a certain degree of risk. The results are usually poorer and more biased, whereas large-sample data tend to be of better quality and less biased. Random effects models may affect the accuracy of the results. Based on this meta-analysis, several diagnostic strategies can be employed to guide clinical practice.

MBI is an inexpensive tracer material that can be easily obtained on the market, and there are no restrictions on its application in hospitals without nuclide equipment, rendering it suitable for promotion in primary hospitals. The MBI staining method has high accuracy in SLNB (Özdemir et al., 2014; Bakhtiar et al., 2016; Wang et al., 2020; Devarakonda et al., 2021; Brahma et al., 2017; Yang et al., 2021). However, the results of this study indicate that it is lower than NCS. A meta-analysis by Li et al. (2018) revealed that mapping the location of sentinel lymph nodes via MB dye alone, as recommended by the American Society of Breast Surgeons, resulted in an acceptable rate of identification but an excessively high rate of false-negatives. Caution is needed when using MB dye alone as a labelling method for sentinel lymph node biopsy.

NCS suspensions are a new type of tracer. The associated imaging principle is that the average diameter of NCS particles is 150 nm, which is larger than the diameter of capillaries (20–50 nm), and they cannot enter capillaries; however, they can smoothly pass through capillary lymphatic vessels (120–500 nm in diameter). When the NCS suspension is injected into the tissues around a tumour, the phagocytosis of macrophages into the capillary lymphatic vessels occurs, and eventually, when the NCS suspension is injected into the surrounding tissues of the tumour, the macrophages phagocytose the charcoal particles into the capillary lymphatic vessels and finally converge into the lymph nodes such that the lymph nodes will be dyed black and, thus, recognized during the operation. Since the hydrostatic pressure in the tissue interstitial space is greater than the pressure in the capillary lymphatic vessels, the NCS will not enter the tissue interstitial space to cause black staining of local tissues. The NCS is relatively inexpensive and does not require the use of special testing equipment. The safety of NCS localization has been confirmed in animal and human experiments (Guo et al., 2018; Wu et al. 2013). Its advantages are fast lymphatic tissue absorption, a long retention time in the SLN, ease of use, affordability, safety and nontoxicity, as well as that it does not cause background staining, unlike water-soluble reagents such as methylene blue (Bianco, Kostarelos & Prato, 2005; Mohajeri, Behnam & Sahebkar, 2018; Loh et al., 2018). The NCS compensates for some of the disadvantages of traditional tracers (Thevarajah, Huston & Simmons, 2005; Chavda et al., 2022; Liu et al., 2017; Gupta et al., 2020; Hermansyah et al., 2021; Somashekhar et al., 2008; Xu et al., 2022), and in the 17 included studies, no allergic reactions, localized inflammatory reactions, or skin and fat necrosis were reported in patients; however, skin staining is more common with the use of the NCS. Five minutes after the administration of NCS, the stained lymph nodes were traced along the black-dyed lymphatic vessels, and the lymph nodes were resected and sent for pathological examination.

Both NCS and MBI have a certain false-negative rate, which may be attributed to the following reasons: First, affected by the operator’s proficiency, subcutaneous injection of tracer may cause tissue necrosis or crusting, or the tracer may penetrate into the tumour; therefore, the operator needs to have a certain degree of clinical experience and familiarity with the operative protocol. Second, tracer success is affected by the exposure of the axilla. Sentinel lymph nodes are usually in a lower position and may exist in several positions or directions simultaneously; thus, if the exposure is not sufficient, they may be incorrectly identified. Third, affected by the location of the tumour, when the tumour is medial, the sentinel lymph nodes may be in the inner breast area, and biopsy of the axilla may result in false-negatives. The efficacy of this method is also influenced by the unique structure of the breast. Scarring of the breast, i.e., disrupted lymphatic ducts or lymphatic vessels with cancerous emboli, results in a change in the course of tracer, making the detection of sentinel lymph nodes erroneous. The results of this study indicate that NCS has a lower false-negative rate compared to MBI, which may be attributed to the smaller particle diameter of NCS, allowing for clearer staining of lymph nodes and lymphatic channels.

There are several limitations of this study. (1) Although an extensive search was conducted for this study, unpublished literature could not be obtained, so potential publication bias cannot be excluded. (2) Further subgroup analyses were not performed due to the limitations of the initial information of the included studies. The quality of the original literature may also affect the results of the meta-analysis. Currently, there are multiple tracers used for SLNB for breast cancer, which have different diagnostic values. In-depth subgroup analysis can provide information that is instructive for diagnosis and identification. (3) Because NCS is a third-generation specific tracer approved by the Chinese Food and Drug Administration, the retrieved studies were all conducted by Chinese scholars, and no studies by scholars from other countries were retrieved for inclusion in the analysis.

Conclusions

We believe that the discriminative ability of NCS is superior to that of MBI and that NCS is a satisfactory tracer in sentinel lymph node biopsy for breast cancer and should be considered as a first-line diagnostic agent for use in patients.

Supplemental Information

Supplemental Information 1 Baseline characteristics and raw data of included studies

Supplemental Information 2 PRISMA checklist

This study was made possible by extensive sample collections initiated by authors; JDF, MH, JS, JRH, SG and the generous donation of specimens by CR Clarke, M Shivji, SA Karl, J Martinez and through collection efforts by the Pacific Islands Regional Observer Program, IATTC Observer Program, American Samoa Regional Observer Program, and Southeast Shark Bottom Longline Observer Program. We thank members of the ToBo Lab for sharing expertise, advice, and discussions that improved this manuscript. We also thank K Freel for providing edits to the manuscript to ensure clarity. We thank the staff of the HIMB EPSCoR Evolutionary Genetics Core Facility and especially A Eggers and M Mizobe for assistance with genotyping. Special thanks to D Lerner, K Holland, C Meyer, D Bethea, D McCauley, and C Wilson for guidance and sage advice. Thanks to the staff at Hawaiʻi Institute of Marine Biology for their support throughout this project. This is contribution #1944 from the Hawaiʻi Institute of Marine Biology and contribution #11767 from the School of Ocean and Earth Science and Technology at the University of Hawaiʻi at Mānoa.

Additional Information and Declarations

Competing Interests

Author Contributions

Data Availability

The authors declare there are no competing interests.

Luhua Xia conceived and designed the experiments, performed the experiments, analyzed the data, prepared figures and/or tables, authored or reviewed drafts of the article, and approved the final draft.

Zuowei Zou performed the experiments, analyzed the data, authored or reviewed drafts of the article, and approved the final draft.

Le Chong conceived and designed the experiments, prepared figures and/or tables, authored or reviewed drafts of the article, and approved the final draft.

Xinhua Wang analyzed the data, authored or reviewed drafts of the article, and approved the final draft.

Zhanfei Dong analyzed the data, authored or reviewed drafts of the article, and approved the final draft.

Yanping Zhao conceived and designed the experiments, analyzed the data, prepared figures and/or tables, authored or reviewed drafts of the article, and approved the final draft.

The following information was supplied regarding data availability:

This is a systematic review/meta-analysis.

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
