# Peer review of "The value of nanocarbon contrast methylene blue based on dye-based tracer technology in sentinel lymph node biopsy for breast cancer: a systematic review and meta-analysis"

_PeerJ, doi:10.7717/peerj.19546_

## Round 0.1 · original submission · Major Revisions

Please pay particular attention to reviewers 2 & 3.

**Language Note:** The review process has identified that the English language must be improved. PeerJ can provide language editing services - please contact us at [email protected] for pricing (be sure to provide your manuscript number and title). Alternatively, you should make your own arrangements to improve the language quality and provide details in your response letter. – PeerJ Staff

Reviewer 1 ·

Basic reporting

The paper is well-written, easy to interpret, and well-structured. The relevant data is included

Experimental design

The research question is well-defined and the methods are well defined

Validity of the findings

Overall. The article is very well written in all its aspects and parts

Additional comments

- The second paragraph in the introduction needs to be referred
- The first paragraph in the discussion can be omitted

Reviewer 2 ·

Basic reporting

The introduction needs to be edit for language, specific details of sections that are difficult to understand are listed below. The discussion is much clearer and easily understood. Subject to language changes, the introduction and background does set the context. Literature references are relevant. Figures are adequate.

Line 42 ‘Systemic metastasis is often judged clinically based on their 44 pathological examination results. Theoretically, if a patient's sentinel lymph nodes do not metastasize, other lymph nodes in the axillary lymphatic drainage area will not metastasize’ – language should be improved – it is the cancer that metastasises, not the sentinel lymph nodes

Line 50 ‘avoids upper limb lymphedema’ – reduces the possibility of

Line 54 ‘The nanocarbon method is used as a tracer’ – abrupt statement

Line 55-60 ‘Methylene blue (MB) is the most commonly used material for the blue dye method. Methylene 56 blue injection (MBI) is less expensive, easy to obtain, less metamorphotic, safe, less toxic, and easy to prepare and perform before surgery, and MBI localization is more accurate and plays an important role in performing minimally invasive biopsy to determine whether the lymph node is metastatic. Methylene blue injection (MBI) is less expensive, easier to obtain, results in fewer allergic reactions, is safe and nontoxic, and is easy to prepare for the procedure and perform’ – this section is repetitive and could be abbreviated

Line 85 ‘The search strategy for Chinese 86 newspapers was similar to that used for English newspapers’ – the word journal should be used rather than newspapers

Line 271 The first part of the article uses the abbreviation NCS, but from line 271 onwards the abbreviation CNS has been used – needs to be standardised

Experimental design

This is a meta-analysis, not primary research. Although the research question has been defined, this metaanalysis has not addressed many other techniques which are available and widely used for sentinel lymph node biopsy. This limits the value of this meta-analysis. Methods described are adequate.

The review limits itself to a comparison of sentinel lymph node mapping using either methylene blue or carbon nanoparticle suspension. There are several otherwidely used techniques for SLNB (radioisotope, magnetic particles, fluorescent mapping) – although some of these are mentioned, the article does not attempt to analyse these techniques. Additionally, apart from methylene blue there are other commonly used blue dyes (eg isosulphan blue, patent blue), which have not been addressed.

Every articles included in the final analsysi is from a Chinese journal. This indicates that the nanoparticle technique does not have wide application in other parts of the world.. Because of this limited approach, although this article is interesting it will may not have wide interest.

Validity of the findings

The impact is likely to be limited, explained above. Underlying data for the selected studies is adequate, although other similar techniques have not been examined. Conclusions are not very clear, as the discussion independently lists the reported benefits or disadvantages of the techniques, but does not provide a direct comparison or explain whether each issue would apply to the other dye.

Additional comments

Line 230 ‘Although the dye method is simple, intuitive and practical, inexpensive, and does not require special equipment, the method has a certain degree of blindness, is time-consuming, and results in greater tissue damage, and the success rate often depends on the choice of a good tracer, the experience of the operator and the accuracy of the pathological examination, in which the choice of a suitable tracer is the first condition’ – this is a circular statement where the sentence starts with the benefits of the dye method, and goes on to state that the success rate depends on the choice of dye

Line 236 ‘three main methods for SLNB’ - other well established techniques should be included in this list,eg fluorescence mapping, magnetic particles – pros and cons need to be discussed.

Paragraph starting with line 271 This describes the mechanism for CNS based sentinel node mapping. The recommended time interval between injection and surgery should be included
Line 292-301 ‘affected by the operator's proficiency, subcutaneous injection of methylene blue may cause tissue necrosis or crusting, or the tracer may penetrate………may result in false-negatives’ - it is likely that CNS would havethe same likelihood of failure as MB – eg false negatives based on the location of axillary lymph nodes, medial tumours, the adequacy of axillary exposure etc. It is also likely that similar clinical expertise is required for accurate CNS based SLN mapping. The authors should analyse the points listed as disadvantages of using MB, and provide an explanation for why these issues would not apply to CNS

Line 295 ‘traumatic cavity or tumour’ – as sentinel lymph node biopsy is specifically related to the management of malignant tumours, the phrase ‘traumatic cavity’ should be omitted.

Line 299 ‘Third, affected by the location of the tumour, when the 300 tumour is medial, the sentinel lymph nodes may be in the inner breast area, and biopsy of the 301 axilla may result in false-negatives’ – this is possible with all techniques for SLNB, and not limited to the methylene blue method

Reviewer 3 ·

Basic reporting

The writing in the article is clear and reads in an unambiguous and professional manner. There is a good number of references that well cover the background information. The structure of the article is appropriate with an introductory section followed by materials and methods, results and discussion. The discussion is informative and explains the usage of methylene blue and carbon nanostructure for sentinel node imaging in good detail. The results described are relevant to the hypothesis that the nanocarbon structures are better for sentinel lymph node imaging than the methylene blue injection method.

Concerning figures, I notice that at the end of the article there are a total of 15 figures presented yet only figures up through 7B are discussed under Results. SROC curves that are key results are Figures 8 and 9 at the end of the paper but SROC curves are referred to as 4A and 4B in the text. There are some issues in the labeling of the figures at the end of the paper: Figure 4 says 'Carbon nanoforest sensitivity' and Figure 5 says 'Carbon nanoscale specificity', the naming needs to be made consistent. Figure 10 says 'Nanocarbon modchk', what is 'modchk'? What is the word 'pubbias' used to label Figures 12 and 13?

Experimental design

The study is a detailed statistical study of results gathered from the literature after a careful screening process that narrows over 200 papers down to 17 studies of sufficient quality. The research question that asks if the nanocarbon suspensions are more reliable than the methylene blue injection method for sentinel lymph node imaging is of high diagnostic importance. The literature results are examined applying appropriate and advanced statistical method. Data extraction and quality assessment with participation of multiple authors suggest a rigorous investigation. The statistical methods are clearly identified so that they could be checked by other interested clinical researchers.

Validity of the findings

The key result is the better SROC curve for nanocarbon suspension compared to that for methylene blue. The difference is small but suggests use of nanocarbon suspension, a more recently introduced and approved method, is certainly giving good results and results seems improved. The data appear to have been reliably extracted from the literature. The conclusions are stated clearly at the end of the paper and also in the abstract.

It would be helpful to clarify the statement 'did not comply with the literature' in section 2.1 under 'Search Results'.

Additional comments

The main issue to be addressed is why are there 13 papers at the end of the paper presented but the discussion goes up to Figure 7B and some of the Figure numbers in the text don't match the figures all at the end of the paper.

The Discussion section at the end is commendable as the details of the methods used for sentinel lymph mode imaging are described in good detail.

---

## Round 0.2 · accepted · Accept

Authors have addressed all of the reviewers' comments and manuscript is ready for the publication.

Reviewer 3 ·

Basic reporting

The revised manuscript is fully improved from the first version. All issues with usage terminology, mostly with respect to the figures and tables, have been resolved. The language is perfectly clear. The structure of the article provides a flow or presentation that is conducive to easy reading. The introduction is well focused on the motivations for the study of comparing methods for labeling sentinel lymph nodes in biopsies related to breast cancer. The focus on evaluating the newer method of using nanocarbon suspensions as compared to methylene blue is important. The raw data for the 17 studies finally selected for analysis are included and analyzed. The results presented lead to a clear conclusion that is stated as NCS has some performance characteristics that are better than those of methylene blue and should be seriously considered as an alternative, newer method.

Experimental design

Yes, the study falls within the scope of the journal. The research question is very well defined: How does the diagnostic value of NCS compare to that of methylene blue for labeling sentinel lymph nodes in biopsies for assessing the metastasis of breast cancer? The standards applied to narrow down the number of studies of sufficient quality for statistical analysis are strict, and a total of 402 identified articles were narrowed down to 17. The statistical methods and software used are identified. The literature databases used are identified. The description of the analysis appears thorough enough for others to repeat if they want. The process of having two literature searchers and an arbiter third person as described, should remove bias and maintain an ethical standard. The knowledge gap filled is the need for a comparison of NCS and methylene blue labeling methods.

Validity of the findings

The conclusion is clearly stated, right at the start of the paper. Subsequent is the analysis and detailed discussion, which also provides a lot of background and insight into the medical procedures related to the two labeling methods. The data extracted from the literature are provided. Since NCS labeling is new, the comparison study has current relevance and novelty.

Additional comments

The paper is much improved in the revision and reads clearly, and the results are presented concisely to reach the conclusion.